# Vaccination for the Prevention of Infection among Immunocompromised Patients: A Concise Review of Recent Systematic Reviews

**DOI:** 10.3390/vaccines10050800

**Published:** 2022-05-18

**Authors:** Kay Choong See

**Affiliations:** Division of Respiratory & Critical Care Medicine, Department of Medicine, National University Hospital, Singapore 119228, Singapore; mdcskc@nus.edu.sg; Tel.: +65-6779-5555

**Keywords:** COVID-19, critical illness, immunocompromised host, influenza, human, sepsis, seroconversion

## Abstract

Vaccination is crucial for avoiding infection-associated morbidity and mortality among immunocompromised patients. However, immunocompromised patients respond less well to vaccinations compared to healthy people, and little is known about the relative efficacy of various vaccines among different immunocompromised states. A total of 54 systematic reviews (22 COVID-19; 32 non-COVID-19) published within the last 5 years in Pubmed^®^ were reviewed. They demonstrated similar patterns within three seroconversion response categories: good (about >60% when compared to healthy controls), intermediate (~40–60%), and poor (about <40%). Good vaccine responses would be expected for patients with chronic kidney disease, human immunodeficiency virus infection (normal CD4 counts), immune-mediated inflammatory diseases, post-splenectomy states, and solid tumors. Intermediate vaccine responses would be expected for patients with anti-cytotoxic T-lymphocyte antigen-4 therapy, hematologic cancer, and human immunodeficiency virus infection (low CD4 counts). Poor vaccine responses would be expected for patients with B-cell-depleting agents (e.g., anti-CD20 therapy), hematopoietic stem-cell transplant, solid organ transplant, and liver cirrhosis. For all vaccine response categories, vaccination should be timed when patients are least immunosuppressed. For the intermediate and poor vaccine response categories, high-dose vaccine, revaccination when patients are less immunosuppressed, checking for seroconversion, additional booster doses, and long-acting monoclonal antibodies may be considered, supplemented by shielding measures.

## 1. Introduction

Immunocompromised patients have weakened immune systems due to chronic illness (e.g., chronic kidney failure) or therapies that depress immunity (e.g., chemotherapy for cancer, immunomodulation for immune-mediated diseases, and anti-rejection drugs for organ transplantation). Consequently, immunocompromised patients suffer increased susceptibility to sepsis. Sepsis, which is the combination of severe infection with a dysregulated response to infection and organ dysfunction [1], is in turn associated with increased morbidity, mortality, and cost of care.

To improve the overall prognosis for immunocompromised patients, both downstream improvements of sepsis care and upstream prevention of infection are crucial. For the latter, vaccination against common pathogens is a key strategy which is recommended by major guidelines [2]. Common vaccine-preventable pathogens include those transmitted via the respiratory route (e.g., SARS-CoV-2, influenza, pneumococcus, and varicella-zoster virus) and those transmitted via other routes (e.g., viral hepatitis A and B, and yellow fever virus). Given the risk of proliferation of attenuated vaccine strains in immunocompromised patients [3], live virus vaccines are contraindicated in patients with active immunosuppression and are only allowed after careful balancing of benefit versus risk [4]. Concerns about vaccine-related relapse of inflammatory rheumatic diseases and post-vaccination allograft rejection exist; however, in general, these appear uncommon [5,6,7,8], and vaccinations should not be withheld on the basis of these concerns [2].

However, just as immunocompromised patients have deficient immunity to defend against infection, such patients may also have deficient immune responses to vaccination, rendering the latter less effective than expected from studies among healthy controls. Substantial reductions in vaccine efficacy (measured within controlled study environments) or effectiveness (measured in real-world studies) using standard vaccination regimes would necessitate enhanced vaccination strategies or the addition of non-vaccine-based preventive methods (e.g., shielding measures).

Systematic reviews on vaccination for the prevention of infection in immunocompromised patients are ideal for aggregating the published literature on individual vaccines and individual immunocompromising conditions. Given that immunocompromised patients are a heterogeneous group with varying levels of immunosuppression, subgroups of immunocompromising conditions with varying vaccine responses may be identified. Knowledge of these patient subgroups may help stratify preventive measures, with more intensive measures being provided for patients with the poorest vaccine response. A review of systematic reviews was, therefore, conducted to elucidate broad immunocompromised patient subgroups. In doing so, this paper can serve as a source of information for readers interested in an overview of studies, as well as stimulate further research into host-dependent classification of vaccine effectiveness.

## 2. Materials and Methods

Using a validated systematic review filter [9], a comprehensive search of Pubmed^®^ (pubmed.ncbi.nlm.nih.gov) was performed (Table 1). To study contemporary and clinically relevant vaccines, the search was limited to papers published within 5 years of 13 April 2022. Studies were excluded if vaccination against infection was not studied, study outcome was not about vaccine efficacy/effectiveness, patients were not immunocompromised, or primary studies were not reviewed.

Screening of titles and abstracts was conducted, and the following data fields were extracted from the full-text documents: vaccine type, number of adult and pediatric patients, number of studies, the reason for being immunocompromised, description of vaccine efficacy, and interventions to improve vaccine efficacy. A qualitative review of included studies was then performed to uncover a general understanding of the associations of various immunocompromising conditions with immune responses to vaccines, as well as construct vaccine seroconversion response categories. In addition, interventions to improve vaccine efficacy were reviewed to inform potential solutions for various vaccine seroconversion response categories.

## 3. Results

Out of 979 studies extracted from Pubmed^®^, 54 systematic reviews (22 COVID-19; 32 non-COVID-19) were included (Table 1).

### 3.1. Systematic Reviews of COVID-19 Vaccines

A total of 22 systematic reviews focused on COVID-19 vaccines (Table 2). Most the COVID-19 vaccines were mRNA-based, while the remainder were viral vector-based and inactivated virus vaccines. Immunocompromised states studied included use of B-cell-depleting anti-CD20 therapy, chronic kidney failure, immune-mediated inflammatory diseases, malignancy, and solid organ transplant recipients.

The COVID-19 vaccine systematic reviews predominantly used seroconversion as a marker for vaccine efficacy. In general, the studies showed that patients with chronic kidney failure on dialysis (not requiring organ transplantation) [10,11,12], immune-mediated inflammatory diseases [13,14,15,16], and solid tumors [17] had seroconversion rates that were high and similar to healthy controls (seroconversion rates among patients ranged from about 83% to 97%). In contrast, patients with receipt of anti-CD20 therapy [18] and solid organ transplants [11,19,20] (thus requiring anti-rejection immunosuppression) had markedly low vaccine seroconversion rates (rates ranged from about 26% to 45% after two vaccine doses). Patients with hematologic cancers had intermediate seroconversion rates (rates ranged from about 54% to 65%) [21,22,23,24,25,26,27,28]. An intervention that improved vaccine efficacy was the use of additional booster vaccine doses [20].

**Table 2 vaccines-10-00800-t002:** Efficacy of COVID-19 vaccination among immunocompromised patients.

Vaccine Type	Author (Year) [Ref]	Number of IC Patients (Studies)	Reason for being IC	Description of Vaccine Efficacy	Interventions to Improve Vaccine Efficacy
COVID-19 (mRNA)	Akyol (2021) [10]	1955 adults (18)	Dialysis and kidney transplant recipients	Pooled seroconversion rate of 27.2% for kidney transplantation, 88.5% for dialysis patients, and 100% for healthy controls after two doses of vaccine	Second vaccine dose
COVID-19 (various)	Becerril-Gaitan (2022) [21]	8332 adults (35)	Malignancy	Pooled seroconversion rate in cancer patients of 51% after first dose vaccine and 73% after second dose vaccine. Seroconversion lower in patients with hematologic cancer versus solid tumors (65% vs. 94%)	Second vaccine dose
COVID-19 (various)	Bhurwal (2022) [13]	2484 adults (21)	Inflammatory bowel disease	Pooled seroconversion rate of 73.7% and 96.8% after one and two doses of vaccine respectively	Second vaccine dose
COVID-19 (mostly mRNA)	Cavanna (2021) [22]	621 adults (6)	Malignancy	No reduced rate of seroconversion for patients with solid tumors compared with the control, but 38% reduced seroconversion for patients with hematologic cancer	Second vaccine dose
COVID-19 (various)	Corti (2022) [23]	9260 adults (36)	Malignancy	Pooled seroconversion rate of 11–87.5% and 7.3–100% after one and two doses of vaccine, respectively. Exceptionally poor seroconversion for patients with hematologic cancer receiving B-cell-depleting agents within last 12 months	Second vaccine dose
COVID-19 (mRNA)	Efros (2022) [19]	853 adults (7)	Solid organ transplant recipients	Pooled seroconversion rate of 50.3% after the third vaccine	Third vaccine dose
COVID-19 (various)	Gagelmann (2021) [24]	11,086 adults (49)	Hematologic malignancies	Pooled seroconversion rate of 64% after 2 doses of mRNA or 1 dose of vector-based vaccines, versus 96% for solid cancer and 98% for healthy controls	Not studied
COVID-19 (various)	Galmiche (2022) [25]	25,209 adults (162)	Malignancy, dialysis, transplant recipients, immune-mediated disease	No seroconversion among 18–100% of solid organ transplant recipients, 14–61% of patients with hematological malignancy, 2–36% of patients with cancer, and 2–30% of patients on dialysis	Not studied
COVID-19 (mRNA, viral vector)	Guven (2021) [27]	1448 adults (17)	Malignancy	Cancer patients had significantly lower seroconversion rates than controls after first vaccine dose (37.3 vs. 74.1%) and after two doses (78.3 vs. 99.6%). The difference in seroconversion rates was more pronounced patients with hematologic cancer versus solid tumors	Second vaccine dose
COVID-19 (mRNA, viral vector)	Guven (2022) [26]	3187 adults (26)	Hematologic malignancies	Pooled seroconversion rate of 33.3% and 65.3% after one and two doses of vaccine, respectively; <70% seroconversion if on anti-CD20 or anti-CTLA-4 therapy	Second vaccine dose
COVID-19 (mRNA)	Jena (2022) [15]	2286 adults (25)	Immune mediated inflammatory diseases	Pooled seroconversion rate 69.3% and 83.1%, after 1 and 2 doses of mRNA vaccination, respectively	Second dose vaccine
COVID-19 (various)	Jena (2022) [14]	9447 adults (46)	Inflammatory bowel disease	Pooled seroconversion rate of 96% for complete vaccination. Decay of titers higher with anti-TNF immunomodulation	Not studied
COVID-19 (94% mRNA)	Lee (2022) [29]	9974 adults (82)	Malignancy, immune-mediated inflammatory disorders, organ transplant recipients, HIV patients	Seroconversion 6–44% after 1 vaccine dose, 35–89% after 2 vaccine doses	Second vaccine dose
COVID-19 (99% mRNA)	Ma (2022) [11]	4264 adults (27)	Chronic kidney failure requiring kidney replacement therapy	After 2 vaccine doses, 44% decreased seropositivity compared to general population. Kidney transplant recipients had significantly lower seroconversion than patients on hemodialysis or peritoneal dialysis (26.1 vs. 84.3% and 92.4%, respectively)	Not studied
COVID-19 (mRNA)	Manothum-metha (2022) [20]	11,713 adults (29)	Solid organ transplant recipients	Mean seroconversion rate was 10.4% after 1 dose, 44.9% after 2 doses, and 63.1% after 3 doses. Lower response given older age, deceased donor status, and use of immunosuppression (antimetabolite, rituximab, and antithymocyte globulin)	Multiple (up to 4) vaccine doses
COVID-19 (various)	Marra (2022) [30]	45,040 adults (24)	Solid organ transplant recipients, malignancy, inflammatory rheumatic diseases	Mean seroconversion rate of 25.2% in solid organ transplant recipients, 68% in patients with malignancy, and 86% in patients with inflammatory rheumatic diseases. Overall vaccine effectiveness of 70.4% against symptomatic infection	Not studied
COVID-19 (various)	Mehrabi (2022) [31]	3207 adults (26)	Autoimmune conditions, malignancy, transplant recipients	A 48% lower rate of seroconversion after 2 doses, worse with transplant recipients	Not studied
COVID-19 (mostly mRNA)	Sakuraba (2022) [17]	1453 adults (16)	Malignancy	Pooled seroconversion rate of 54.2% and 87.7% after one and two doses of vaccine, respectively; lower rates with hematologic compared to solid organ malignancy	Second vaccine dose
COVID-19 (mostly mRNA)	Sakuraba (2022) [16]	5360 adults (25)	Immune-mediated inflammatory diseases	Pooled seroconversion rate of 73.2% and 83.4% after one and two doses of vaccine, respectively; lower rates with patients on anti-CD20 therapy	Second vaccine dose
COVID-19 (various)	Schietzel (2022) [18]	1342 adults (23)	Anti-CD20 therapy	Pooled seroconversion rate of 40% for complete vaccination; lower rates with kidney transplant recipients	Not studied
COVID-19 (various)	Swai (2022) [12]	2789 adults (27)	Chronic kidney failure	Hemodialysis patients’ proportions of humoral (antibody) and cellular (T-helper cell) immune responses varied from 87.3% to 88.8% and from 62.9% to 85.8%, respectively, comparable to healthy control responses. Kidney transplant patients’ humoral and cellular immune responses ranged from 2.6% to 29.9% and from 5.1% to 59.8%, respectively, significantly lower than healthy control responses	Not studied
COVID-19 (various)	Teh (2022) [28]	7064 adults (44)	Hematologic malignancies	Overall seroconversion rate 37–51% after 1 dose of COVID-19 vaccine and 62–66% after 2 doses	Second vaccine dose

CTLA-4: cytotoxic T-lymphocyte antigen-4; HIV: human immunodeficiency virus; IC: immunocompromised; mRNA: messenger ribonucleic acid; Ref: reference; TNF: tumor necrosis factor. Various vaccine types included mRNA, viral vector-based, and inactivated virus vaccines.

### 3.2. Systematic Reviews of Non-COVID-19 Vaccines

A total of 32 systematic reviews focused on non-COVID-19 vaccines (Table 3). Several vaccines were studied, including inactivated hepatitis A vaccine [32], recombinant hepatitis B vaccine [33,34,35,36,37,38], recombinant human papillomavirus vaccines [39,40], inactivated influenza vaccine [41,42,43,44,45,46,47,48,49,50,51], live-attenuated measles vaccine (given post hematopoietic stem-cell transplantation [52] and in children living with human immunodeficiency virus [53]), pneumococcal conjugate and polysaccharide vaccines [46,49,54,55,56], live-attenuated and recombinant subunit zoster vaccines [57,58], live-attenuated yellow fever vaccine [59], and others [60,61,62,63]. Immunocompromised states studied included use of B-cell-depleting anti-CD20 therapy, chronic kidney failure, human immunodeficiency virus infection, immune-mediated inflammatory diseases, liver cirrhosis, malignancy, post-splenectomy status, and solid organ transplant recipients.

The non-COVID-19 vaccine systematic reviews predominantly used seroconversion as a marker for vaccine efficacy. In general, the studies showed that patients with immune-mediated inflammatory diseases [33,34,38,43,46,47,49,61], human immunodeficiency virus infection [35,37,39,40], post-splenectomy status [55], and solid tumors [48] had seroconversion rates that were high and similar to healthy controls (seroconversion rates among patients ranged from about 61% to 100%). In contrast, patients with receipt of anti-CD20 therapy [50], solid organ transplants [32,42], and liver cirrhosis [36] had markedly low vaccine seroconversion rates (rates ranged from about 3% to 49%). Patients with hematologic cancers [58] had intermediate seroconversion rates (about 52%). Systematic reviews involving children [35,39,40,42,44,52,53,54,59,60,61,62,63] demonstrated findings consistent with those involving adults. An intervention that improved vaccine efficacy was the use of high-dose vaccines [35,37,42,44,45,51].

## 4. Discussion

### 4.1. Overview of Results

According to the humoral responses to vaccines of various immunocompromising conditions, the systematic reviews for COVID-19 and non-COVID-19 vaccines demonstrated similar patterns. Given the absence of any validated guidelines or recommendations for classification of vaccine seroconversion, three seroconversion response categories were arbitrarily constructed from the data: good (about >60% when compared to healthy controls), intermediate (~40–60%), and poor (about <40%) (Table 4). Good vaccine responses would be expected for patients with chronic kidney disease, human immunodeficiency virus infection with normal CD4 counts, immune-mediated inflammatory diseases, post-splenectomy states, and solid tumors. Intermediate vaccine responses would be expected for patients with anti-cytotoxic T-lymphocyte antigen-4 therapy, hematologic cancer, and human immunodeficiency virus infection with low CD4 counts. Poor vaccine responses would be expected for patients with B-cell-depleting agents (e.g., anti-CD20 therapy), hematopoietic stem-cell transplant, solid organ transplant, and liver cirrhosis.

For all vaccine response categories, vaccination should be timed when patients are least immunosuppressed (e.g., before initiating immunosuppressive treatment) and when immunosuppressive disease states are optimally treated [64]. For the intermediate and poor vaccine response categories, methods to improve vaccine response include the use of high-dose vaccine and revaccination when patients are less immunosuppressed. These vaccine-based methods should also be supplemented by non-vaccine methods such as shielding measures (e.g., face mask use, hand hygiene, and physical distancing for respiratory pathogens). For the poor vaccine response category, given possible vaccine nonresponse, seroconversion may be checked. If nonresponse is demonstrated, additional booster doses may be considered [11,20,65]. Alternatively, long-acting monoclonal antibodies for pre-exposure prophylaxis may be considered in patients at high risk of acquiring serious infection [66].

### 4.2. Limitations of the Current Study

Firstly, this review only analyzed systematic reviews included in Pubmed^®^, which would not encompass systematic reviews included in other databases such as Embase©. Nonetheless, the individual systematic reviews would have included papers from both Pubmed^®^-listed and non-Pubmed^®^-listed journals, and missing important data are, hence, unlikely. Secondly, this review was limited to the last 5 years of publication, which would exclude older systematic reviews covering other vaccines. However, this would avoid the inclusion of noncontemporary vaccines, which might affect the overall pattern of vaccine efficacy among immunocompromised patients. Thirdly, this review did not include a quality assessment of individual systematic reviews, as it was the intention to be as inclusive as possible. Nevertheless, the studies all showed fairly consistent results, and stratification by study quality would not have affected the overall interpretation of the current findings. Fourthly, given the heterogeneity of studies, statistical pooling of the results could not be performed, although the current results may serve as a source of information for readers interested in an overview of studies. Lastly, systematic reviews were not available for studying the interaction of immune-mediated diseases and many immunosuppressive medications or for studying combinations of immunocompromised states. Logically, the vaccine response is expected to worsen with greater doses or number of immunosuppressive medications and with coexisting immunocompromised states.

### 4.3. Limitations of the Included Systematic Reviews

Although a substantial number of non-COVID-19 vaccine systematic reviews contained data from children [35,39,40,42,44,52,53,54,59,60,61,62,63] (13 out of 32), none of the COVID-19 vaccine systematic reviews included pediatric studies. Furthermore, none of the systematic reviews had long-term vaccine efficacy or effectiveness results, and none were performed for oral vaccines (e.g., oral rotavirus, oral cholera, oral polio, and oral typhoid). These important knowledge gaps could potentially be filled once more primary studies are available.

### 4.4. Future Directions

Vaccine development is often a long process involving multiple rounds of preclinical studies and clinical trials. Even for clinically successful vaccines, eventual vaccine efficacy would be dependent on both pathogen and host [67,68]. From this study, according to the systematic reviews focusing on vaccines targeting a variety of pathogens, the type of immunosuppression in the host appears to play an important role. This review provides a broad overview of various vaccine studies, leading to the construction of three seroconversion response categories. Further studies, which include prediction of vaccination efficacy using baseline measures of circulating B cells [69], could be performed to refine these categories and to highlight exceptions within these categories. These categories should also be validated against long-term serological protection and clinical effectiveness data. In addition, interventions to improve vaccine efficacy are limited and more studies are required to investigate novel methods such as heterologous prime-boost techniques [70] and long-acting preventive antibody therapy [66].

## 5. Conclusions

In conclusion, this review of 54 systematic reviews demonstrated three vaccine seroconversion response categories among immunocompromised patients: good (about >60% when compared to healthy controls), intermediate (~40–60%), and poor (about <40%). For all vaccine response categories, vaccination should be timed when patients are least immunosuppressed. For the intermediate and poor vaccine response categories, high-dose vaccine, revaccination when patients are less immunosuppressed, checking for seroconversion, additional booster doses, and long-acting monoclonal antibodies may be considered, supplemented by shielding measures.

## Figures and Tables

**Table 1 vaccines-10-00800-t001:** Study selection and reasons for exclusion.

Study Inclusion/Exclusion Criteria	Number of Studies
Pubmed search performed on 13 April 2022, for articles published within the last 5 years. Search terms: (“vaccine” [MeSH] OR “vaccination” [MeSH] OR vaccine [All Fields] OR vaccination [All Fields] OR vaccine* [All Fields]) AND (“cancer” [MeSH] OR “cancer” OR “malignancy” OR “malign*” OR “immunocompromise” [All Fields] OR “immunocompromised” [All Fields] OR “immunodeficiency” [All Fields] OR “immunodeficient” [All Fields] OR “immunodef*” [All Fields] OR “immunocompr*” [All Fields] OR “chemotherapy” [All Fields] OR “chemo*” [All Fields] OR “immunosuppressed” [All Fields] OR “immunosuppression” [All Fields] OR “immunosupp*” [All Fields] OR “rheumatology” [All Fields] OR “rheumatic” [All Fields] OR “rheum*” [All Fields] OR “autoimmune” [All Fields] OR “autoimmunity” [All Fields] OR “transplant” [All Fields] OR “solid organ” [All Fields] OR steroids [MeSH] OR antineoplastic agents [MeSH] OR chemotherapy [MeSH] OR cytotoxicity [MeSH] OR immunologic [MeSH] OR antirheumatic agents [MeSH] OR immunosuppressive agents [MeSH] or steroid* or corticosteroid* or (antineoplastic* AND agent*) OR chemotherap* or cytotoxic*) AND ((“Meta-Analysis as Topic” [MeSH] OR meta analy* [TIAB] OR metaanaly* [TIAB] OR “Meta-Analysis” [PT] OR “Systematic Review” [PT] OR “Systematic Reviews as Topic” [MeSH] OR systematic review* [TIAB] OR systematic overview* [TIAB] OR “Review Literature as Topic” [MeSH]) OR (cochrane [TIAB] OR embase [TIAB] OR psychlit [TIAB] OR psyclit [TIAB] OR psychinfo [TIAB] OR psycinfo [TIAB] OR cinahl [TIAB] OR cinhal [TIAB] OR “science citation index” [TIAB] OR bids [TIAB] OR cancerlit [TIAB]) OR (reference list* [TIAB] OR bibliograph* [TIAB] OR hand-search* [TIAB] OR “relevant journals” [TIAB] OR manual search* [TIAB]) OR ((“selection criteria” [TIAB] OR “data extraction” [TIAB]) AND “Review” [PT])) NOT (“Comment” [PT] OR “Letter” [PT] OR “Editorial” [PT] OR (“Animals” [MeSH] NOT (“Animals” [MeSH] AND “Humans” [MeSH])))	979
Exclusion 1: Vaccination against infection not studied	590
Exclusion 2: Study outcome was not about vaccine efficacy/effectiveness	170
Exclusion 3: Patients were not immunocompromised	120
Exclusion 4: Primary studies were not systematically reviewed	45
Included in final review	54

**Table 3 vaccines-10-00800-t003:** Efficacy of non-COVID-19 vaccination among immunocompromised patients.

Vaccine Type	Author (Year) [Ref]	Number of IC Patients (Studies)	Reason for Being IC	Description of Vaccine Efficacy	Interventions to Improve Vaccine Efficacy
HAV	Garcia (2019) [32]	1332 adults (17)	Organ transplant recipients, or chronic inflammatory conditions	In organ transplant recipients, seroconversion was 0–67% after 1 dose and 0–97% after 2 doses. In patients with chronic inflammatory conditions, seroconversion was 6–100% after 1 dose and 48–100% after 2 doses	Second vaccine dose
HBV	Jiang (2017) [33]	1688 adults (13)	Inflammatory bowel disease	Pooled seroconversion response rate 61%, better for younger patients, when vaccinated during disease remission or when not on immunosuppression	Vaccination ideally done before starting immune-suppression or during disease remission
HBV	Kochhar (2021) [34]	2375 adults (14)	Inflammatory bowel disease	Pooled proportion of adequate immune response 64%, with 87% reduced odds of response vs. healthy controls	Not studied
HBV	Lee (2020) [35]	914 adults, 56 children (9)	HIV	99% increased odds of vaccine seroconversion response among patients who received a double (40 mcg) vs. standard (20 mcg) dose	Double better than single dose
HBV	Rodrigues (2019) [36]	Not stated (24)	Liver disease requiring transplant	Seroconversion 20–54% for cirrhotic patients vs. 74–97% for healthy controls. No seroconversion superiority between the accelerated (3 months) vs. conventional (6 months) immunization schedules	Accelerated immunization schedules
HBV	Tian (2021) [37]	1821 adults (17)	HIV	Pooled seroconversion rate of 71.5%, worse with lower CD4 counts	Double better than single dose. Four-dose schedule better than three-dose schedule
HBV	Singh (2022) [38]	2602 adults (21)	Inflammatory bowel disease	62% adequate immune response (>10 IU/L) and 42% effective immune response (>100 IU/L), worsened with immunosuppression	Not studied
HPV	Mavundza (2020) [39]	916 adults, 126 children (5)	HIV	96–100% seroconversion rate for both bivalent and quadrivalent vaccines	Not studied
HPV	Zizza (2021) [40]	950 adults/children (4)	HIV	~100% seroconversion rate for all 3 vaccines studied	Not studied
Influenza	Bitterman (2018) [41]	2275 adults (6)	Cancer	Generally decreased mortality, influenza-like illness and pneumonia	Adjuvanted vaccine (inconclusive)
Influenza	Chong (2018) [42]	943 adults/children (7)	Organ transplant recipients	Seroconversion (7–63.9%) rates for influenza antigens were low and not improved by intradermal or adjuvanted influenza vaccines. Some benefit from high dose (60 mcg) vaccine and booster doses, compared to standard single dose (15 mcg)	Alternative vaccine strategies (intradermal; adjuvanted; high-dose; booster doses)
Influenza	Huang (2017) [43]	886 adults (13)	Rheumatoid arthritis	Similar seroconversion compared to healthy controls, and 44–49% lower for patients receiving non-steroid immunosuppression	Adjuvanted vaccine more immunogenic than non-adjuvanted
Influenza (trivalent)	Lai (2019) [44]	888 adults, 132 children (8)	Transplant or chemo-therapy recipients	High-dose vaccine (60 mcg) increased seroconversion over standard dose (15 mcg) by 13% for A/H1N1 strains, and was well-tolerated	High vs. standard dose
Influenza (trivalent)	Leibovici (2021) [45]	41,313 adults (3)	Older adults ≥65 years and IC	24% decreased risk of laboratory-confirmed influenza for high-dose (60 mcg) vs. low-dose (15 mcg) vaccine	High dose (4×) trivalent vaccine
Influenza	Muller (2022) [46]	644 adults (7)	Inflammatory bowel disease	No differences in antibody responses were observed compared to non-immunosuppressed patients	Not studied
Influenza	Nguyen (2021) [47]	332 adults (9)	Multiple sclerosis on treatment	No difference in antibody response in multiple sclerosis patients vs. healthy controls	Not studied
Influenza	Spagnolo (2021) [48]	986 adults (8)	Immune checkpoint inhibitor therapy for cancer	Seroconversion rate 52–65% from one study. No differences in terms of antibody titers compared with healthy age-matched controls from another study	Not studied
Influenza	Subesinghe (2018) [49]	350 adults (7)	Rheumatoid arthritis	Influenza vaccine humoral responses preserved with methotrexate and tumor necrosis factor inhibitor exposure. Combined methotrexate and tocilizumab/tofacitinib associated with reduced pneumococcal and influenza vaccine humoral responses	Not studied
Influenza	Vijenthira (2021) [50]	222 adults (13)	Anti-CD20 therapy	Pooled seroconversion 3% after 1 pandemic influenza vaccine dose, 95% less than healthy controls	Not studied
Influenza	Zhang (2018) [51]	2015 adults (13)	HIV	Adjuvanted 7.5 mcg booster and 60 mcg single vaccine strategies provided 2–3 times better seroconversion and seroprotection outcomes, than single 15 mcg vaccine	High dose (4×) vaccine; adjuvanted vaccine
Measles (live attenuated)	Groeneweg (2021) [52]	319 children (10)	Transplant recipients	Seroconversion rates 41–100% after 1 dose and 73–100% after 2 doses in solid organ transplant recipients; 33–100% after 1 dose and 100% after 2 doses in hematopoietic stem-cell transplant recipients	Not studied
Measles	Mehtani (2019) [53]	335 children (12)	HIV	30–56% decreased seroconversion in HIV-infected children vs. uninfected controls	Not studied
Pneumococcal	Adawi (2019) [54]	571 adults, 30 children (18)	SLE	Protective antibody titers 36–97.6%, lower with high erythrocyte sedimentation rate, older age, earlier SLE onset, high disease activity, and immunosuppressive therapy	Not studied
Pneumococcal	Lenzing (2022) [55]	2430 adults (21)	Post-splenectomy	No differences in antibody responses were observed compared to healthy controls	Optimal vaccination timing but results were uncertain
Pneumococcal	Muller (2022) [46]	785 adults (5)	Inflammatory bowel disease	No decreased seropositivity with non-anti-TNF therapy and 72% decreased seropositivity with anti-TNF therapy, compared to non-immunosuppressed patients	Not studied
Pneumococcal	Subesinghe (2018) [49]	141 adults (2)	Rheumatoid arthritis	Methotrexate associated with reduced pneumococcal vaccine humoral response. Combined methotrexate and tocilizumab/tofacitinib associated with reduced pneumococcal and influenza vaccine humoral responses	Not studied
Pneumococcal	van Aalst (2018) [56]	1623 adults (22)	Autoimmune disease receiving immuno-suppression	After PCV vaccination, 26% seroconversion among patients vs. 47% among controls. After PPSV, seroconversion 37% among patients vs. 50% among controls	Not studied
VZV (ZVL or VZVsub)	Hamad (2021) [57]	404,561 adults (8)	Chronic kidney failure including transplant patients	45% lower risk of VZV in vaccinated patients vs. controls, but no effect for the subgroup of transplant patients	Not studied
VZV (VZVsub)	Racine (2020) [58]	1389 adults (6)	Various patient groups ^e^	Humoral immune response 50–93%; 52% for patients with hematological malignancy. Vaccine efficacy against infection was 67–72% in hematopoietic stem-cell transplant patients	Not studied
Yellow fever (live attenuated)	Martin (2021) [59]	561 adults, 18 children (10)	HIV	97.6% seroconversion	Not studied
Various ^a^	Adeto-kunboh (2019) [60]	66,220 children (14)	HIV	Pneumococcal vaccine efficacy (to prevent disease) 32% for HIV-infected vs. 78% for HIV-uninfected. BCG protection 0% for HIV-infected vs. 59% for HIV-uninfected. Hib protection 44% for HIV-infected vs. 97% for HIV-uninfected. Rotavirus vaccines similar protection for HIV-infected and HIV-uninfected	Not studied
Various ^b^	Dembinski (2020) [61]	1130 patients, 10–24 years (20)	Inflammatory bowel disease	Immunogenicity of vaccinations not altered by IBD status or use of immunosuppression	Not studied
Various ^c^	Keller (2022) [62]	2571 children (37)	Juvenile autoimmune rheumatic diseases on immune-suppressive treatment	9 studies showed seroconversion rates lower in children with juvenile autoimmune rheumatic diseases on immune-suppressive treatment compared with control children, but many other studies were underpowered to demonstrate differences	Not studied
Various ^d,e^	Tse (2020) [63]	2468 children (37)	Children with autoimmune diseases on immune modulatory drug therapy	Patients on biologics mounted adequate seroprotective responses, but antibody titers tended to be lower. Patients on steroids had decreased seroconversion (60% for influenza vaccine). Patients on cyclophosphamide had decreased seroconversion (50% for HPV vaccine)	Not studied

^a^ Included 9-valent pneumococcal conjugate vaccine, BCG, Hib; ^b^ included diphtheria, HAV, HBV, HPV, influenza, pertussis, pneumococcus, and VZV vaccines; ^c^ included diphtheria, HAV, HBV, HPV, influenza, measles, mumps, meningococcal C, pneumococcus, rubella, tetanus, and VZV vaccines; ^d^ included HAV, HBV, HPV, influenza, measles, mumps, rubella, tetanus, and VZV vaccines; ^e^ included patients with autologous stem-cell transplant, hematological cancer, HIV infection, renal transplant, and solid tumor under chemotherapy. BCG: Bacillus Calmette–Guérin; HAV: hepatitis A virus; HBV: hepatitis B virus; Hib: *Haemophilus influenzae* type b; HIV: human immunodeficiency virus; HPV: human papillomavirus; IC: immunocompromised; PCV: pneumococcal conjugate vaccine; PPSV: pneumococcal polysaccharide vaccine; Ref: reference; SLE: systemic lupus erythematosus; TNF: tumor necrosis factor; VZV: varicella-zoster virus; ZVL, live-attenuated zoster vaccine; VZVsub, recombinant subunit zoster vaccine.

**Table 4 vaccines-10-00800-t004:** Vaccine seroconversion response categories.

Category	General Seroconversion Rates	IC Types	Suggested Management
Good response	About >60% compared to healthy controls	Chronic kidney disease requiring hemodialysis or peritoneal dialysis HIV (normal CD4 counts) Immune-mediated inflammatory diseases (e.g., RA, SLE) Inflammatory bowel disease Multiple sclerosis (treated) Post-splenectomy status Solid tumors	Follow usual vaccination regime (including any booster doses), by default Time vaccination when least immunosuppressed
Intermediate response	About 40–60% compared to healthy controls	Anti-CTLA-4 therapy Hematologic cancer HIV (low CD4 counts)	Time vaccination when least immunosuppressed Shielding measures ^a^ Consider high-dose vaccine, revaccination when less immunosuppressed
Poor response	About <40% compared to healthy controls	B-cell-depleting agents (e.g., anti-CD20 therapy) Hematopoietic stem-cell transplant recipients Liver cirrhosis Solid organ transplant recipient	Time vaccination when least immunosuppressed Shielding measures ^a^ Consider high-dose vaccine, revaccination when less immunosuppressed Consider checking seroconversion. If nonresponse, consider booster doses or long-acting monoclonal antibodies for pre-exposure prophylaxis ^b^

^a^ Examples include face mask use, hand hygiene, and physical distancing for respiratory pathogens; ^b^ examples include tixagevimab–cilgavimab for COVID-19. CTLA-4: cytotoxic T-lymphocyte antigen-4; HIV: human immunodeficiency virus; IC: immunocompromised; RA: rheumatoid arthritis; SLE: systemic lupus erythematosus.

## Data Availability

Data supporting this review are available in the reference section.

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
