# Peer review of "Vaccination for the Prevention of Infection among Immunocompromised Patients: A Concise Review of Recent Systematic Reviews"

_vaccines, 2022, doi:10.3390/vaccines10050800_

Round 1
Reviewer 1 Report
This paper represents a review of other reviews. It does not report findings according to PRISMA guidelines, nor does it include pooling of results, but it does represent a relatively comprehensive search of recent relevant systematic reviews. However, some shortcomings include:
- The outcome(s) – whether seroconversion or infections - are not always clearly stated (as an example, see ref 24, Marra).
- Lack of mention of any adverse effects from vaccination in immunocompromised recipients, and no mention of vaccine contraindications.
- Given it is a review of reviews without any attempt at pooling the results, it lacks novelty, but may nevertheless serve as a reasonable source for anyone interested in an overview of studies.
Minor comment: I did only a quick search and noted that this article was not included: https://pubmed.ncbi.nlm.nih.gov/33507119/
Author Response
Thank you for your time and comments. I have written my point-by-point replies below.
Comment 1: This paper represents a review of other reviews. It does not report findings according to PRISMA guidelines, nor does it include pooling of results, but it does represent a relatively comprehensive search of recent relevant systematic reviews. However, some shortcomings include:
The outcome(s) – whether seroconversion or infections - are not always clearly stated (as an example, see ref 24, Marra).
Reply 1: Thank you for spotting the omission; the seroconversion outcomes for Marra (2022) have been added. In addition, the outcomes for Galmiche (2022), Jiang (2017), Lee (2020), Tian (2021), Chong (2018), Huang (2017), Nguyen (2021), and Subesinghe (2018) have been clarified.
Comment 2: Lack of mention of any adverse effects from vaccination in immunocompromised recipients, and no mention of vaccine contraindications.
Reply 2: These points have been added to the Introduction: “Given the risk of proliferation of attenuated vaccine strains in immunocompromised patients (Schrauder et al., 2007), live virus vaccines are contraindicated in patients with active immunosuppression, and are only allowed after careful balancing of benefit versus risk (Huber et al., 2018). Concerns about vaccine-related relapse of inflammatory rheumatic diseases and post-vaccination allograft rejection exist, though in general these appear uncommon (Del Bello et al., 2021; Dos Santos et al., 2016; Nakafero et al., 2019; Stevens et al., 2020), and vaccinations should not be withheld based on these concerns (Rubin et al., 2014).”
Comment 3: Given it is a review of reviews without any attempt at pooling the results, it lacks novelty, but may nevertheless serve as a reasonable source for anyone interested in an overview of studies.
Reply 3: This limitation has been added to the Discussion: “Fourthly, given the heterogeneity of studies, statistical pooling of the results could not be done, though the current results may serve as a source of information for readers interested in an overview of studies.”
Comment 4: Minor comment: I did only a quick search and noted that this article was not included: https://pubmed.ncbi.nlm.nih.gov/33507119/
Reply 4: Thanks for pointing out this omission. This paper (Tse, Borrow, & Arkwright, 2020) was part of the papers extracted using the search strategy but was erroneously excluded. It has been added to the review.
Reviewer 2 Report
This reviewer feels that the information from this review article is not a MUST read in the vaccine research area. Although the summary is within the scope of this journal, this reviewer does not understand the motivation of this paper. Developing an effective vaccine for a specific parasite/pathogen usually takes more than ten years and many rounds of clinical trials. The logical design and post-production of an antigen are not always successful and should be considered both host- and pathogen-dependent. As also pointed out by the author, the limitation of this review is obvious and more attention should be paid to the section on future perspectives. Moreover, the introduction section does not make any sense and no references have been cited and thus must be recognized properly.
Author Response
Thank you for your time and comments. I have written my point-by-point replies below.
Comment 1: This reviewer feels that the information from this review article is not a MUST read in the vaccine research area. Although the summary is within the scope of this journal, this reviewer does not understand the motivation of this paper.
Reply 1: To clarify the motivation of this paper, the following statement has been included in the Introduction: “In doing so, this paper can serve as a source of information for readers interested in an overview of studies, and to stimulate further research into host-dependent classification of vaccine effectiveness.”
Comment 2: Developing an effective vaccine for a specific parasite/pathogen usually takes more than ten years and many rounds of clinical trials. The logical design and post-production of an antigen are not always successful and should be considered both host- and pathogen-dependent. As also pointed out by the author, the limitation of this review is obvious and more attention should be paid to the section on future perspectives.
Reply 2: Added to the section on future perspectives: “Vaccine development is often a long process involving multiple rounds of pre-clinical studies and clinical trials. Even for clinically successful vaccines, eventual vaccine efficacy would be dependent on both pathogen and host (Dhakal & Klein, 2019; Langwig et al., 2019). From this study, according to the systematic reviews focusing on vaccines targeting a variety of pathogens, the type of immunosuppression in the host appears to play an important role.”
Comment 3: Moreover, the introduction section does not make any sense and no references have been cited and thus must be recognized properly.
Reply 3: The Introduction has been improved in the following areas:
- Included the full definition of sepsis and included its citation: “Sepsis, which is the combination of severe infection with a dysregulated response to infection and organ dysfunction (Singer et al., 2016), is in turn associated with increased morbidity, mortality and cost of care.”
- Included the citation for the 2013 IDSA Clinical Practice Guideline for Vaccination of the Immunocompromised Host: “For the latter, vaccination against common pathogens is a key strategy which is recommended by major guidelines (Rubin et al., 2014).”
- To clarify the motivation of this paper, the following statement has been included in the Introduction: “In doing so, this paper can serve as a source of information for readers interested in an overview of studies, and to stimulate further research into host-dependent classification of vaccine effectiveness.”
Reviewer 3 Report
The concise review on the preventive effect of SARS-CoV-2 vaccines in immune compromised individuals has a clear aim of the study, its methods are well described and the author uses an easily understandable language. The paper is well structured and its findings are reported in a straightforward manner.
There is some redundancy in the description of results, which ist due to the way how findings are structured: in part 3.1 / Table 2 only results of COVID-19-vaccines are shown, in part 3.2 / Table 3 only results of Non-COVID-19 vaccines. In both chapters (p. 3 of 15, lines 90-95; p. 6 of 15, lines 128-132) the author uses response categories which are defined only later in the text (discussion, p. 10 of 15, lines 166-167).
I therefore suggest that the author gives exact numberts/ranges of seroconversion rates in the results' section (instead of using phrases like "high and similar to healthy controls", "markedly low" and "intermediate").
Minor point: p. 6 of 15, line 118 should probably read "living with human immunodeficiency virus".
Author Response
Thank you for your time and comments. I have written my point-by-point replies below.
Comment 1: The concise review on the preventive effect of SARS-CoV-2 vaccines in immune compromised individuals has a clear aim of the study, its methods are well described and the author uses an easily understandable language. The paper is well structured and its findings are reported in a straightforward manner.
There is some redundancy in the description of results, which ist due to the way how findings are structured: in part 3.1 / Table 2 only results of COVID-19-vaccines are shown, in part 3.2 / Table 3 only results of Non-COVID-19 vaccines. In both chapters (p. 3 of 15, lines 90-95; p. 6 of 15, lines 128-132) the author uses response categories which are defined only later in the text (discussion, p. 10 of 15, lines 166-167).
I therefore suggest that the author gives exact numberts/ranges of seroconversion rates in the results' section (instead of using phrases like "high and similar to healthy controls", "markedly low" and "intermediate").
Reply 1: Thank you for your comment. I have included the required numerical data in the results section (underlined), and included liver cirrhosis in the poor response category instead of the intermediate response category after reviewing the data:
- For COVID-19 vaccines: “In general, the studies showed that patients with chronic kidney failure on dialysis (not requiring organ transplantation) (Akyol et al., 2021; Ma et al., 2022; Swai et al., 2022), immune-mediated inflammatory diseases (Bhurwal et al., 2022; Jena, James, et al., 2022; Jena, Mishra, et al., 2022; Sakuraba, Luna, & Micic, 2022b) and solid tumors (Sakuraba, Luna, & Micic, 2022a) had seroconversion rates that were high and similar to healthy controls (seroconversion rates among patients ranged from about 83% to 97%). In contrast, patients with receipt of anti-CD20 therapy (Schietzel et al., 2022) and solid organ transplants (Efros et al., 2022; Ma et al., 2022; Manothummetha et al., 2022) (hence requiring anti-rejection immunosuppression) had markedly low vaccine seroconversion rates (rates ranged from about 26% to 45% after 2 vaccine doses). Patients with hematologic cancers had intermediate seroconversion rates (rates ranged from about 54% to 65%) (Becerril-Gaitan et al., 2022; Cavanna, Citterio, & Toscani, 2021; Corti et al., 2022; Gagelmann et al., 2021; Galmiche et al., 2022; Guven, Sahin, Akin, & Uckun, 2022; Guven, Sahin, Kilickap, & Uckun, 2021; Teh et al., 2022).”
- For non-COVID-19 vaccines: “In general, the studies showed that patients with immune-mediated inflammatory diseases (Dembinski, Dziekiewicz, & Banaszkiewicz, 2020; Huang, Wang, & Tam, 2017; Jiang et al., 2017; Kochhar et al., 2021; Muller et al., 2022; Nguyen, Hardigan, Kesselman, & Demory Beckler, 2021; Singh et al., 2022; Subesinghe, Bechman, Rutherford, Goldblatt, & Galloway, 2018), human immunodeficiency virus infection (Lee et al., 2020; Mavundza, Wiyeh, Mahasha, Halle-Ekane, & Wiysonge, 2020; Tian et al., 2021; Zizza et al., 2021), post-splenectomy status (Lenzing, Rezahosseini, Burgdorf, Nielsen, & Harboe, 2022) and solid tumors (Spagnolo et al., 2021) had seroconversion rates that were high and similar to healthy controls (seroconversion rates among patients ranged from about 61% to 100%). In contrast, patients with receipt of anti-CD20 therapy (Vijenthira, Gong, Betschel, Cheung, & Hicks, 2021), solid organ transplants (Chong, Handler, & Weber, 2018; Garcia Garrido et al., 2019) and liver cirrhosis (Rodrigues, Silva, Felicio, & Silva, 2019) had markedly low vaccine seroconversion rates (rates ranged from about 3% to 49%). Patients with hematologic cancers (Racine et al., 2020) had intermediate seroconversion rates (about 52%).”
In addition, the seroconversion response category thresholds were corrected from <30%/40-60%/>70% to <40%/40-60%/>60%, as the former categorization was discontinuous.
Comment 2: Minor point: p. 6 of 15, line 118 should probably read "living with human immunodeficiency virus".
Reply 2: Thank you for spotting the error. The phrase has been corrected.
Round 2
Reviewer 2 Report
This reviewer has no further comments on this revised manuscript and recommends accepting this review paper.